# Relationships between Cognitive Functioning and Powered Mobility Device Use: A Scoping Review

**DOI:** 10.3390/ijerph182312467

**Published:** 2021-11-26

**Authors:** Alice Pellichero, Lisa K. Kenyon, Krista L. Best, Marie-Eve Lamontagne, Marie Denise Lavoie, Éric Sorita, François Routhier

**Affiliations:** 1Department of Rehabilitation, Université Laval, Quebec City, QC G1V 0A6, Canada; alice.pellichero.1@ulaval.ca (A.P.); marie-eve.lamontagne@fmed.ulaval.ca (M.-E.L.); francois.routhier@rea.ulaval.ca (F.R.); 2Centre for Inter-Disciplinary Research in Rehabilitation and Social Integration (Cirris), Centre Intégré Universitaire de Santé et de Services Sociaux de la Capitale-Nationale (CIUSSS-CN), Quebec City, QC G1M 2S8, Canada; 3Department of Physical Therapy, Grand Valley State University, Grand Rapids, MI 49401, USA; kenyonli@gvsu.edu; 4Université Laval-Bibliothèque, Quebec City, QC G1V 0A6, Canada; marie-denise.lavoie@bibl.ulaval.ca; 5Université de Bordeaux—Handicap Activité Cognition Santé (EA 4136 HACS), 33076 Bordeaux, France; eric.sorita@chu-bordeaux.fr

**Keywords:** cognitive functioning, power mobility devices, evaluations, training

## Abstract

Background. Powered mobility devices (PMD) promote independence, social participation, and quality of life for individuals with mobility limitations. However, some individuals would benefit from PMD, but may be precluded access. This is particularly true for those with cognitive impairments who may be perceived as unsafe and unable to use a PMD. This study explored the relationships between cognitive functioning and PMD use. The objectives were to identify cognitive functions necessary to use a PMD and describe available PMD training approaches. Methods. A scoping review was undertaken. Results. Seventeen studies were included. Four examined the predictive or correlational relationships between cognitive functioning and PMD use outcomes with intellectual functions, visual and visuospatial perception, attention, abstraction, judgement, organization and planning, problem solving, and memory identified as having a relation with PMD use outcome in at least one study. Thirteen others studied the influence of PMD provision or training on users’ PMD capacity and cognitive outcomes and reported significative improvements of PMD capacities after PMD training. Six studies found improved cognitive scores after PMD training. Conclusions. Cognitive functioning is required to use a PMD. Individuals with heterogeneous cognitive impairment can improve their PMD capacities. Results contribute to advancing knowledge for PMD provision.

## 1. Introduction

Mobility is recognized as a basic human right [1]. For individuals with mobility limitations, powered mobility devices (PMD), such as powered wheelchairs and scooters, may provide independent mobility which may not be otherwise possible [2]. The prevalence of PMD use has increased exponentially during the last few decades in industrialized countries. Globally, in the United States of America, Canada and the United Kingdom, the number of PMD users has multiplied by three times in the past 20 years [3,4,5]. With an aging population, the prevalence of PMD use will continue to increase.

For individuals who benefit from PMD, while the mode of transportation may change, the importance remains constant; transportation from one location to another is critical to engagement in meaningful activities. For example, among community-dwelling older adults, PMD use is associated with an increased frequency of grocery shopping and going for “walks”, and an increased frequency of instrumental activities of daily living, such as going to a restaurant, posting letters, going to the bank, and visiting family and friends [6]. In children, PMD use contributed to the development of cognitive and play skills [7] while increasing independence and social interactions [8].

However, many individuals who might benefit from PMD may be excluded access due to substantial challenges with cognition and memory (e.g., due to the natural aging processes, intellectual impairment, brain damage, stroke) [9]. In clinical and community-based practice, for individuals who have learning challenges (including those with cognitive impairment), several evaluations and training sessions may be required to meet the needs of PMD users. Moreover, subjective clinical judgement often plays a central role in determining whether an individual has the necessary cognitive functions for using a PMD [10]. Clinicians reported cognitive functioning as their top concern when providing a PMD [11]. Hence, individuals with dual cognitive and mobility impairments may be precluded from PMD provision before they get a chance to benefit from training [11], resulting in missed opportunities for occupational engagement and social participation.

Global association between cognitive functioning and PMD use has been demonstrated [12]. However, specific cognitive functions required for PMD use remain unclear. Therefore, decisions around PMD provision are commonly based on subjective representations that may preclude some individuals from access to a PMD. For this reason, restricted mobility may be an issue for some individuals with dual cognitive and mobility impairments who have restricted access to assistive technologies.

A scoping review exploring the relationships between cognitive functioning and PMD use may contribute to a fundamental advancement of knowledge to ensure best practices for PMD provision. For the purposes of this study, cognitive functioning refers to a construct within ‘Body functions’, in the mental functions chapter of the International Classification of Functioning (ICF) [13] and PMD use was defined as representing both ‘capacity’ (i.e., what a person can do in a standard environment) and ‘performance’ (i.e., what a person actually does in their everyday environment [13]). Such a review may contribute to evidence-based knowledge for clinicians and could enhance PMD provision for individuals with dual cognitive and mobility impairments.

The aim of this scoping review was to explore the relationships between cognitive functioning and PMD use among PMD users with dual cognitive and mobility impairment. Specific objectives were (1) to identify assessments used to evaluate cognition in relation to PMD use, (2) to identify cognitive functions necessary to use a PMD, (3) to describe PMD training approaches.

## 2. Materials and Methods

### 2.1. Study Design and Registration

A scoping review was conducted to explore the nature and the extent of research evidence [14,15] and to clarify concepts related to cognitive function and PMD use. The intent of the research was to describe the scope of current evidence [16] including all study designs and all ages of power wheelchair users. Findings from this scoping review may orient research question development and selection of inclusion criteria for a future systematic review [15,16].

To ensure rigor this scoping review adhered to the ‘Preferred Reporting Items for Systematic Reviews and Meta-Analyses (PRISMA) statement’. The study protocol was registered a priori with the International Prospective Register of systematic reviews, (PROSPERO CRD42019118957) and a protocol was published [17].

### 2.2. Data Sources and Searches

A search strategy was developed with the support of a research librarian (MDL). Appropriate key words were selected according to Subject Headings, terms used in existing studies on cognition and PMD use, and the “Mental functions” chapter of the ICF. The search strategy generated 147 terms related to cognitive functioning. The search was conducted in online databases including MEDLINE (Ovid), CINAHL (Elsevier), PsycINFO (Ovid) and Web of Science (Clarivate). Search strategies are available in the Appendix A. The search was performed from the inception of each database in February 2019 and was updated in March 2020. Reference lists of the included studies were hand searched to ensure coverage of the literature available.

### 2.3. Study Selection

Inclusion and exclusion criteria: Scientific peer-reviewed studies including PMD users (inclusive of ages and diagnoses), reporting cognitive functioning and PMD use (capacity and/or performance) outcomes, presenting original data, and published in English or French, were included. Editorial, commentaries and theoretical papers were excluded.

Selection procedure: Identified studies were exported to Covidence systematic review software (Veritas Health Innovation, Melbourne, Australia), where duplicates were removed automatically based on the title of the references. Remaining duplicates were deleted during the abstract and title screening. Eligibility was determined through abstract and title reading and by reading the full-text articles. Screening and eligibility were carried out independently by two authors (A.P. and L.K.), and discrepancies were resolved by a third reviewer (K.B.).

### 2.4. Appraisal of the Methodological Quality of Included Studies

The studies were appraised by design in descending order from the highest level of evidence to the lowest level according to the Oxford Center for Evidence-Based Medicine 2011 Levels of Evidence. The methodological quality of each study was appraised using the Mixed Methods Appraisal Tool (MMAT, 2018 version. Registration of Copyright (#1148552), Canadian Intellectual Property Office, Industry, Canada) [18]. The MMAT is a critical appraisal tool designed for appraisal in reviews that include diverse methodologies (qualitative, randomized controlled trials, non-randomized studies, quantitative descriptive studies, and mixed-methods studies). An algorithm allows choice of the appropriate category of study to appraise. For each study, two screening questions and five criteria were scored ‘yes’, ‘no’ or ‘can’t tell’. In this review, MMAT scores 5 and 4 were considered as high methodological quality, and ≤3 were considered as fair methodological quality. Methodological limitations identified in primary studies were considered in the synthesis and interpretation of results. Methodological quality evaluation was completed by one author (A.P.) and discussions with another author (K.B.).

### 2.5. Data Extraction and Synthesis

Data were extracted independently by one reviewer (A.P.) into study-specific extraction tables. The same data extraction approach was applied across all the studies following a standard data extraction template, but with flexibility according to various methodologies.

The studies were organized according to their purpose and by age (children, adults, older adults). Studies were organized by group. The first group included studies evaluating the predictive or correlational associations between PMD use and cognitive outcomes; cognitive assessments tools were summarized. The cognitive functions identified in each outcome measure were classified according to the ICF [13]. The items of each outcome measure were examined to determine which cognitive functions were assessed. The second group explored the effects of PMD provision or training on PMD use and cognitive outcomes. The review team leaders (A.P., K.B., E.S.) engaged in multiple discussion throughout the analysis to ensure that reliability, trustworthiness, and consensus were reached.

## 3. Results

The initial search resulted in 4253 titles (search in 2019 *n* = 3968; updated search in 2020 added *n* = 285). After removal of duplicates, 3027 titles and abstracts were screened, and 126 full texts were reviewed, 17 studies met the inclusion criteria (PRIMSA flowchart in Figure 1).

Data characteristics, levels of evidence and methodological quality of the studies included are presented in Table 1.

The sample size across studies had a median of *n* = 6.5 participants for a total of *n* = 278 participants. Participants ranged in age from 7 months to 81 years. Ten studies included infants, children and adolescents (1–17 years) [7,8,19,20,21,22,23,24,25,26], four studies included adults (18–59 years) [27,28,29,30], and three studies included older adults (>60 years) [31,32,33].

### 3.1. Studies Looking for Predictive or Correlational Associations between PMD Mobility and Cognitive Outcomes (n = 4)

Four studies looked specifically for the associations between PMD mobility and cognitive outcomes [20,26,29,31]. Within these four studies, thirteen outcome measures were used to assess cognitive functioning.

#### 3.1.1. Children

Furumasu et al. (2004) [20] and Tefft et al. (1999) [26] examined the predictive power of cognitive functions required to perform tasks (basic and overall) using a PMD. Furumasu et al. (2004) [20] used a pre-post study (*n* = 50) and Tefft et al. (1999) [26] conducted a cross-sectional study (*n* = 26) (Level III of evidence, high methodological quality). Furumasu et al. (2004) [20] and Tefft et al. (1999) [26] used both the Pediatric Powered Wheelchair Screening Test (PPWST) to assess cognitive functioning and the Powered mobility program evaluation to assess PMD capacity. Both authors reported that the PPWST (evaluating visuospatial perception, organization and planning and problem-solving) predicted a large amount of the variance in assistance required to perform tasks using a PMD (74% and 57%, respectively).

#### 3.1.2. Adults

Massengale et al. (2005) [29] used a cross-sectional design (*n* = 62) (Level III of evidence, high methodological quality). The authors found statistically significant positive correlations between performance scores on the Power Mobility Road Test (PMRT) and (1) outcomes of the Motor free Visual Perception Test-Revised (MVPT-R) (evaluating visual perception), (2) outcomes of the Test of Nonverbal Intelligence—3rd edition (TONI-3rd) (evaluating abstraction, problem solving and intellectual functions), (3) outcomes of the Wechsler Adults Intelligence Scale—Revised (WAIS-R, evaluating judgement, sustaining attention and short-term memory).

#### 3.1.3. Older Adults

Cullen et al. (2008) [31] investigated whether psychological variables were prospectively predictive of PMD use using pre-post study (*n* = 81) (Level III of evidence, high methodological quality). A wide range of outcome measures were used to explore which cognitive outcome predicted PMD indoor and outdoor use. Outcomes of the Repeatable Battery for the Assessment of Neuropsychological Status (R-BANS) (item delayed story recall, evaluating memory functions) were found to predict indoor PMD frequency use. Authors also combined cognitive variables to calculate a mean z score based on the sample distribution to serve as an index of global cognitive impairment, which appeared to negatively influence indoor PMD frequency use. Only long-term memory appeared to predict outdoor frequency of PMD use, accounting for 35% of the variance.

Table 2 (‘Classification of the outcome measures (and items) related to cognition used in the studies according to the ICF’) presents detailed information for each measure and classification of items according to the ICF.

### 3.2. Studies Exploring the Effects of a PMD Provision or Training on PMD Mobility and Cognitive Outcomes (n = 13)

#### 3.2.1. Studies in Children

Jones et al. (2012) [7] examined the effects of a one-year PMD intervention for young children with severe disabilities (Level II evidence, high methodological quality). PMD were provided to the intervention group. During the intervention, parents were asked to provide daily opportunities for their children to sit in the PMD and to encourage the child to experiment with movements while avoiding telling the child what to do. Written guidelines were provided to the parents. This intervention demonstrated an increase with large effect sizes in PMD capacity scores (assessed using the Pediatric Evaluation of Disability Inventory (PEDI), mobility items) when compared to the control group (no PMD provision). No difference was found in cognitive scores assessed using the PEDI (cognitive items) and the Battelle Developmental Inventory (communication and cognition items).

Bottos et al. (2001) [8] used a pre-post design to investigate the effects of early PMD with 29 children with tetraplegia (Level III evidence, high methodological quality). Six to eight months after PMD provision, an increase in PMD capacity, assessed with a modified version of the Powered mobility program evaluation, was found. No difference was found in cognitive outcomes (assessed using Intellectual Quotient (IQ)).

Butler et al. (1984) [19] conducted a retrospective exploratory cross-sectional study to explore whether young children with severe disabilities could learn to drive a PMD (Level III evidence, fair methodological quality). PMDs were introduced at home by parents. During a 4-month observation period, parents had to encourage the child to sit in the PMD for several hours a day, to give them the opportunity to experiment with the PMD in open spaces, to permit supervised play and to respect resistance to engagement in further activity. Authors reported that even if children obtained equal cognitive scores (assessed by clinical observations and the Developmental Profile II), there were disparities related to the duration required to learn how to drive a PMD.

Kenyon et al. (2018) [23] used a A-B-A-B single subject research design to evaluate the influence of an individualized PMD intervention on a child having severe disabilities (Level IV evidence, high methodological quality). The individualized intervention (two intervention periods, each one time per week for 45–60 min for 4 weeks) consisted of the development of basic power mobility skills in engaging environments. Results reported an improvement (standard error measure) in PMD capacity scores as assessed using the Wheelchair Skills Checklist, the Assessment of Learning Power mobility and the PEDI-computer-adaptative-test (PEDI-CAT) (mobility items). No major change was found in cognitive scores as assessed using the Dimension of Mastery Questionnaire and the PEDI-CAT (cognitive items).

Jones et al. (2003) [21] conducted a case report to explore improvements in a 20-month-old child with spinal muscular atrophy through PMD training (Level IV evidence, high methodological quality). The intervention (6-weeks) was based on motor learning principles, included daily opportunities to use the PMD and verbal encouragements from adults. Results described a positive trend in mobility scores (assessed using the PEDI, mobility items) and in cognitive scores (assessed using the PEDI (cognitive items), and by the Battle Developmental Inventory) after the intervention.

Kenyon et al. (2017) [22] conducted a case series evaluating the impact of PMD training on three children (1–3 years old) with severe multiple impairment (Level IV evidence, high methodological quality). Individualized goals related to PMD capacity were determined; practice with the PMD took place within an individually engaging environment, once per week (60 min) for 12 weeks. After the intervention, results described significant improvements in mobility scores for the two oldest children using the PEDI-CAT (mobility items), and for the three participants using the Assessment of Learning Powered mobility. Results also described significant improvements in cognitive scores assessed using the PEDI (cognitive items) for the three children.

Nilsson et al. (2003) [25] conducted a cases report discussing the effects of an intensive joy-stick-operated training in two children (4 and 5 years old) with profound cognitive impairments (Level IV evidence, low methodological quality). The training used manual guidance, hand-over-hand assistance, and verbal feedback, and was realized one to three times a week (30–90 min). After about 4 months, the training was transferred to the children’s homes where the parents and assistants carried it out for 8 months. After the intervention results described positive behavioral changes among the two children, the girl “was able to carry out self-initiated driving” and the boy “occasionally (..) managed to perform self-initiated driving”. The two children displayed increased wakefulness and alertness, the beginning of goal-directed hand use, and an incipient sense of the simple relationship between their action on the joystick and the motion of the chair

Lynch et al. (2009) [24] conducted a case report evaluating the feasibility of providing PMD training opportunities to a 7-month-old child with spina bifida (Level IV evidence, fair methodological quality). Directional training trials (drive to retrieve a toy) and open explored periods were practiced with infant-friendly training, 3 to 4 times per week for 5 months. After the intervention results described a positive trend in mobility scores (number of joystick activations, path length, percentage directed driving success). Results also described improvements in cognitive scores assessed using the Bailey III (cognition, language reception and language expression).

#### 3.2.2. Studies in Adults

Mountain et al. (2014) [30] conducted a randomized controlled trial (RCT) to evaluate the effect of the Wheelchair Skills Training Program (WSTP) version 4.1 in individuals after a stroke (Level III evidence, high methodological quality). Five one-on-one training sessions of 30 min (3 to 5 sessions per week) were provided to the intervention group. The control group received no training. The intervention group demonstrated an increase in PMD capacity scores (assessing PMD capacities using the Wheelchair Skills Test, WST) when compared to the control group. Participants in the intervention group had a higher cognitive level than participants in the control group (on average 26 in the Montreal Cognitive Assessment (MoCA) test vs 20, *p* = 0.02). Of note, even with higher cognitive levels, the intervention group did not use their PMD more frequently (hours per day) at baseline compared to the control group.

Kenyon et al. (2015) [28] conducted a case report describing outcomes of using a PMD with a young adult (18 years old) with severe impairments (Level IV evidence, high methodological quality). An intervention based on practicing PMD skills and self-exploration with meaningful activities was provided for twelve weeks (60 min, twice a week). Results described after-training improvements in mobility scores (assessed using the Power Mobility Screen, mobility items) and in cognitive scores (assessed using the Power Mobility Screen, cognitive items).

Benford et al. (2017) [27] conducted a case report to explore the experience of switch operated PMD training for one young adult with multiple learning disabilities (Level IV evidence, fair methodological quality). The Driving to Learn intervention [42] was provided for 23 weeks (31 sessions of 30–45 min) of. The results described after-training improvements in mobility scores (assessed using the Assessment of Learning Powered Mobility). Qualitative data described that the participant’s energy level influenced PMD capacity and that PMD capacity enhanced the participant’s mood.

#### 3.2.3. Studies in Older Adults

Mountain et al. (2010) [33] conducted a pre-post study with older adults with stroke, with and without unilateral neglect (Level III evidence, high methodological quality). Both groups (with and without unilateral neglect) were provided with the Wheelchair skills training program (WSTP) version 3.2 (5 sessions of 30 min). Both groups demonstrated an increase in PMD capacity scores (assessed using the Wheelchair Skills Test). There was no difference in the extent of improvement between individuals with or without neglect.

Dawson and Thronton (2003) [32] used a A-B-A single subject experimental design to evaluate whether three individuals with unilateral neglect after a stroke could improve their PMD capacity (Level IV evidence, fait methodological quality). A two-week intervention (30 min every weekday) based on a right hemisphere activation approach was provided. All three participants had improved their mobility scores (measured by the number of collisions recorded and time taken negotiating an obstacle) but their visual neglect did not change.

## 4. Discussion

This scoping review explored the relationship between cognitive functioning and PMD use. Research aiming to surpass mobility limitations for people with dual cognitive and motor impairments is increasingly important due to the ageing population [43]. Seventeen studies met the inclusion criteria. However, including multiple moderate-quality studies, this scoping review provides only a moderate strength of evidence and highlights that this area of research has not been deeply investigated, as previously mentioned [44]. Among the seventeen studies, samples of children, adults and older adults were represented, allowing the consideration of cognitive functioning in relation to PMD use through a large age range. However, as cognitive resources and data processing are different for children, adults and older adults, further research is necessary to determine relationships between cognitive functioning and PMD use through the lifespan. Only three studies included older adults, which is in contrast with the fact that research aiming to surpass mobility limitations for older adults is essential due to the aging population [45]. Moreover, only studies including power wheelchair users were found; no studies included scooter users. Given the prevalence of scooter users is almost three times higher than powered wheelchair users [4] this is surprising.

### 4.1. Studies Looking for the Predictive or Correlational Associations between PMD Mobility and Cognitive Outcomes

Results from the four studies looking for the predictive or correlational associations between PMD mobility and cognitive outcomes are in agreement with previous research looking for factors associated with PMD driving [12,46]. Studies on children found that visuospatial perception and problem-solving predicted assistance required to perform tasks using a PMD [20,26]. Comparatively, as overall cognitive impairment worsened, indoor PMD frequency use decreased among older adults [29], which is supported by previous findings demonstrating that the odds of PMD proficiency increased by 7% on average with increased cognition [12]. In older adults, better long-term memory also predicted an increased frequency of indoor and outdoor PMD use [31]. Long-term memory was assessed by Cullen et al. (2008) [31] through delayed story recall, which captured the success of three basic processes: encoding the story, storage, and retrieval. One possible explanation may be that long-term memory troubles occurred via perception and working memory difficulties, as recently demonstrated [47]. In the case of Cullen et al. (2008) [31], it could be hypothesized that working memory (belonging to executive functioning) predicted a non-negligeable part of frequency of indoor and outdoor PMD use, as reported by Massengale et al. (2005) [29], Furumasu et al. (2004) [20] and Tefft et al. (1999) [26].

Largely, results from the four studies looking for the predictive or correlational associations between PMD mobility and cognitive outcomes suggest that intellectual functions, visual and visuospatial perception, attention, abstraction, judgement, organization and planning, problem solving, and memory have a statistically significant relationship with PMD use (at least once, in at least one study). In the context of driving a car, the cognitive functions recommended for evaluation in people with cognitive impairment who want to drive after a brain injury include reaction time, attention functions, visuospatial capacities, executive functions, and memory functions [48]. These cognitive functions identified as fundamental to driving a car resonate with those highlighted as important for PMD use.

### 4.2. Studies Exploring the Effects of a PMD Provision or Training on PMD Mobility and Cognitive Outcomes

Results from the studies exploring the effects of a PMD provision or training on PMD mobility and cognitive outcomes identified diverse PMD training programs, from standardized training approaches to fully individualized interventions. Despite the diversity of the training provided, the thirteen studies reported improvements in PMD capacity scores after training. This agrees with previous research highlighting that individuals with diverse cognitive impairments receiving PMD training can improve their PMD capacities [42]. However, evaluations to measure PMD capacities varied through the studies. Thus, effects of the PMD training approaches used are not comparable. Research specifically comparing PMD intervention effects should be investigated in adults, as it has previously been for children [44].

Related to cognitive scores, improvements in these were not expected after training aiming to improve PMD capacity and performance. Surprisingly considering this, six studies reported improvements in cognitive and mobility outcomes [21,22,24,25,27,28]. Those six studies were case report designs and realized individualized training in children and young adults. Due to their study designs, it cannot be established whether improvements in cognitive scores were due to PMD use, the individualized training, cognitive maturation, or other factors that may not have been controlled. Publishing case reports with favorable results implies publication bias, as previously identified [44]. Other research, conducted by Nilsson between 1999 and 2019 on training young children with profound cognitive disabilities to drive a PMD, showed that mobility experiences promoted increased wakefulness, curiosity, and understanding of cause–effect [49]. Accordingly, the ‘Driving to learn program’ has been developed using a PMD as a therapeutic tool rather than as a means of mobility [9]. Another explanation could be that PMD provision or training may facilitate cognitive improvement through self-generated mobility and environmental exploration, as previously demonstrated in children [50].

Among the seven studies that reported improvement in mobility scores only [7,8,19,23,30,32,33] two, providing PMD to young children (2 and 6 years old), reported that cognitive scores did not improve after 6 months provision or 1 year training. This is surprising in young children as cognitive improvements were expected due to cognitive maturation alone. This may be partially explained by the measures used to assess cognition, which may not have been sensitive to changes in cognitive level. In an RCT evaluating the effect of powered wheelchair skills training on wheelchair skills capacity in adults after a stroke, Mountain et al. [30] affirmed that individuals with stroke could improve PMD capacity. However, the control group had a statistically significant lower cognitive level compared to the intervention group. Therefore, it is not evident if improvements in wheelchair skills capacity was due to the training or were induced by differences in cognitive levels.

Given that learning is dependent on cognitive level [51] and that the learning process requires time and efforts and does not guarantee the quality of knowledges stored [52], the control group may have experienced restriction in their personal experiences with PMD based on the type of training provided (e.g., customization to their needs). Therefore, participants in the control group had lower cognitive levels to start with and were not given the opportunity to be trained. Moreover, given that there were differences in baseline cognition between groups, but no differences in PMD frequency use (hours per day), findings suggest that for individuals with stroke frequency of use was not dependent on presence or absence of cognitive impairment. However, there was no indication if participants were driven independently of their PMD by engaging in meaningful occupations or if they were assisted by others. Thus, in further research the number of hours spent in the PMD per day should not be considered a sole indicator for PMD use. It could thus be hypothesized that frequency of PMD use may mediate the relationship between cognition and driving performance.

Jones et al. conducted two studies examining the influence of PMD provision and training on development and function of young children, which had contradictory results [7,21]. The first was a case report in 2003, which suggested that a young child learnt to maneuver a PMD within a few weeks and improved cognitive functioning. In contrast, the RCT conducted in 2012 suggested that children required at least 12 weeks of intervention, reporting that 7 of the 11 children had not mastered all PMD skills by the end of 1 year, and that their cognitive scores did not change. Jones et al. explained that variability in their findings may be related to the broad inclusion criteria, as the sample included children with a large range of sensorimotor and cognitive impairments compared to the previous reports [7,21]. The variability in impairments may have contributed to the variability in time required to acquire the PMD skills and the lack of change in cognitive scores, as relationship between cognitive resources and learning quality has previously been demonstrated [51].

Results from the studies exploring the effects of PMD provision or training on PMD mobility and cognitive outcomes suggests that, even without cognitive outcome measure changes, improvement in PMD capacity scores demonstrated that despite cognitive limitations participants acquired new abilities facilitating success across mobility domains. This process clearly highlights the participants’ learning potential [53].

### 4.3. Implications

Despite high variability in study designs and publication dates, it appears fundamental to consider cognitive functioning when providing a PMD. Although there have been changes in clinical practice and evolution of PMDs, the role of cognitive functioning is still important for PMD use. While it is true that some intelligent PMDs may play a role in reducing cognitive load for some individuals, artificial intelligence cannot assume all responsibility for the person. Regardless, considering cognitive functioning alone should not be the deciding factor for PMD provision as it is not sufficient to capture the complex interactions between the person, the environment, and the occupation through PMD use. Combining appropriate cognitive assessments and real-life situations would help clinicians to apprehend functional cognitive abilities and secured practices, as indicated to assess car driving abilities after a traumatic brain injury [54]. Moreover, three populations with cognitive impairments can be better described when considering PMD provision: (1) people who objectively can be identified as cognitively able to use a PMD (without or with mild cognitive impairment), (2) people who cannot be identified as cognitively able to use a PMD (with major cognitive impairment), and (3) people for which it is premature to affirm that they can use a PMD or for which it would be unfair to affirm that they cannot. Evaluating objectively this third population and proposing adapted training would offer more individuals a chance to learn to use a PMD. Innovative training approaches customized to individuals with cognitive impairment could, for example, include practicing meaningful activities with the PMD and be conducted in realistic environments and contexts for the user. Meaningful activities should be client-centered and driven largely by client goals. This is in accordance with the philosophy of the ‘Driving to learn program’ [42]. Such practices may reduce existing occupational injustices and improve mobility and quality of life for individuals with dual motor and cognitive impairment.

Exploring the relationship between PMD use and cognitive functioning should also consider PMD technological evolutions. The studies included were conducted between 1984 and 2017 and PMD have greatly changed over time. Still, cognitive functioning implied in PMD use remains the same. Moreover, such research may orient the development of intelligent PMDs which have a similar rationale, providing mobility options for people with dual motor and cognitive impairment [55,56]. For example, intelligent PMDs may enhance secure driving (i.e., automatic stop) and can facilitate some complex maneuvering (i.e., enter in an elevator). While there may be added benefits to PMD users and caregivers (e.g., less worry and risk of accidents), controlled mobility can limit the individual’s ability to make choices about their participation in meaningful activities. Properly programing an intelligent PMD to the users’ cognitive abilities could enhance mobility without restricting autonomy and choice [11].

### 4.4. Limitations

While comprehensive in search, this scoping review has some limitations. First, it is possible that the combination of keywords may have missed some studies despite consulting with experts on the keyword choice. Moreover, data extraction was not conducted by two separate individuals in duplicate. However, transparency was enhanced by regular discussions and team meetings, presentation of emerging findings, and sharing of findings with authors. All steps and decisions were recorded in a logbook. Second, in relation to the evolution of PMD and clinical practice, the large range in publication dates represents a limitation. Although studies reported on small sample sizes (thus limiting generalizability), the search and synthesis spanned all age groups. Despite differences in cognitive impairment and age between participants, this broad synthesis of a heterogenous group was justified at this early stage in research on PMD and cognition to provide a comprehensive overview of existing literature. The studies were organized by age (children, adults, elderly), as children are developing cognitively, adults have already developed their cognition, and cognition among the elderly may be declining, in order to aid in interpretation of the findings. Finally, the ICF framework was used to classify cognitive functions assessed in existing outcome tools based on the content assessed by each tool. Using the ICF allowed a common language, comprehension, and clear definitions. However, application of the ICF framework was subjective, such that other researchers may have classified content differently. An interdisciplinary team with expertise in cognition and PMD use were involved in classification and interpterion of the findings.

## 5. Conclusions

This scoping review supports a relationship between cognitive functioning and PMD use and provides a first step in orienting clinicians and researchers to cognitive considerations for PMD use. Findings suggest that cognitive functioning should not be overlooked when providing PMD. However, given that individuals with heterogeneous cognitive impairment can improve their PMD capacities, cognitive impairment should not preclude PMD provision. To improve mobility and quality of life for individuals with dual motor and cognitive impairment, innovative and customized training approaches should be developed. Such training approaches could include practice of client-centered goals related to using a PMD and be conducted in the users’ actual environment.

There is high variability in the designs, outcome tools and PMD training used throughout the studies identified and heterogenous samples were investigated. Thus, the level and quality of evidence was low. Further studies are needed to understand and model the links between cognitive functioning and PMD use. Such studies are required to explore how cognitive functioning and psychological factors, such as confidence, influence PMD use, and to document which cognitive evaluations could be used with individuals with dual motor and cognitive impairment in clinical practices.

## Figures and Tables

**Figure 1 ijerph-18-12467-f001:**
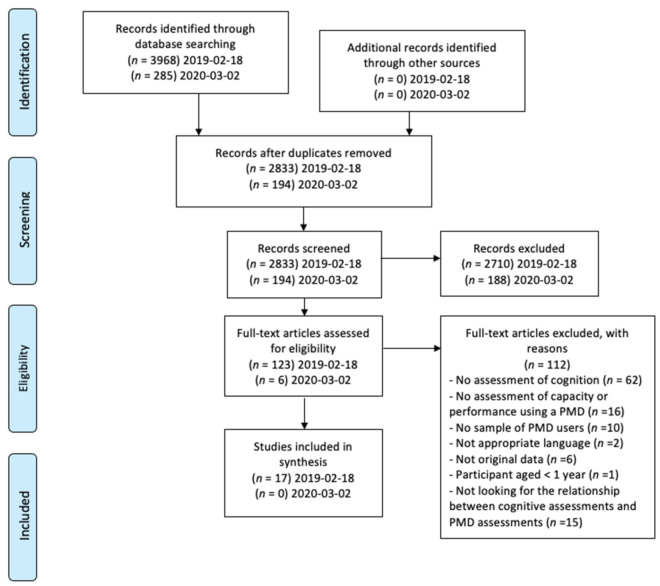
PRIMSA flowchart.

**Table 1 ijerph-18-12467-t001:** Data characteristics, levels of evidence and methodological quality of the studies included (*n* = 17).

1st Author, Publication Date	Time of Assessments	Sample Sample Size, Mean Age, (Range), Diagnosis	Intervention	Outcome Measurement (Items)	Results	Level of Evidence MMAT Score
**Randomized Controlled Trials (*n* = 2)**						
Jones et al., 2012	Baseline, Post PMD provision and training (12 months)	*n* = 28 IG, *n* = 14 21 months (14–30 months) CG, *n* = 14 22 months (14–30 months) cerebral palsy, diverse myopathy, arthrogryposis, progeria, tetraphocomelia, failure to thrive, myotonic dystrophy, hydrocephalus, Dandy Walker syndrome, Achondroplastic dwarfism	IG PMD provision 12-months period Parents had to provide daily opportunities to sit in the PMD, to encourage the child to experiment with movements and to avoid telling the child what to do Written guidelines were provided CG No PMD provision, no training	cognition (a) PEDI (cognitive) (b) Battelle Developmental Inventory (b1) communication (b2) cognitive PMD capacity (a) PEDI (mobility)	cognition (a) *p* = 0.38 (b) (b1) communication *p* = 0.42 (b2) cognitive *p* = 0.38 PMD capacity (a) *p* = 0.02 *	II 5/5
Mountain et al., 2014	Baseline, Post training	*n* = 17 IG, *n* = 9 58.7 years (n.r.) CG, *n* = 8 49.0 years (n.r.) stroke	IG Wheelchair skills training program, version 4.1 5 one-on-one training sessions 30 min each CG No training	cognition (baseline) (a) MoCA, (b) Behavioural Inattention Test, (c) Praxis PMD capacity (a) WST (baseline and follow up)	cognition (baseline) (a) *p* = 0.027 * (b) *p* = 0.533 (c) *p* = 0.619 PMD capacity (a) baseline *p* = 0.924 follow-up *p* = 0.006 *	II 4/5
**Pre-Post Studies (*n* = 3)**						
Bottos et al., 2001	Baseline, Post PMD provision	*n* = 29 6.3 years (3–8) tetraplegia	Provision of a PMD (6 to 8 months)	cognition (a) IQ (performance) (b) IQ (verbal) capacity (a) Powered mobility program evaluation	cognition (a) no significant change (b) no significant change capacity (a) *p* < 0.01 *	III 4/5
Cullen et al., 2008	Pre-post study	*n* = 81 65.6 (29–96) diverse diagnosis (e.g., arthritis, multiple sclerosis, stroke, Parkinson disease…)	1-month PMD provision	cognition (a) IQ (b) Addenbrooke’s Cognitive Examination-R (naming and comprehension) (c) RBANS (verbal memory, visual memory, visual perception) (d) F-A-S Test (e) Behavioural Inattention Test (line bisection) (f) BADS (key search subtest) (g) Depression Anxiety Stress Scale-21 (h) Visual Object and Space perception Battery (screening test, position discrimination, cube analysis) (i) Road Map Test of Direction Sense test PMD capacity (a) self-rated power wheelchair use questionnaire: (level of functional powerchair use (less frequent = use < 3 days per week and <2 activities; for outdoor use < 2 outdoor activities) PMD performance (a) Functional Evaluation in a Wheelchair	cognition/capacity More frequent indoor power wheelchair use was predicted by better score on: - index of overall impairment (zβ = −0.495, *p* = 0.021 *) - verbal memory (zβ = 0.614, *p* < 0.001 *), More frequent outdoor power wheelchair use was predicted by betterscore on delayed story recall (zβ = 0.610, *p* = 0.001 *) PMD performance (a) median score = 89%	
Furumasu et al., 2004	Pre-post study	*n* = 50 n.r. (21 months—6 years 11 months) *n* = 26 triplegic or tetraplegic cerebral palsy, *n* = 24 orthopedic or neuromuscular disabilities	Powered mobility program: motivational play and exploration six sessions 1 h 1 month period	cognition (a) PPWST (problem solving, spatial relations) (b) Symbolic Representation Scale PMD capacity (a) Powered mobility program evaluation	cognition/capacity (a) both factors in the PPWST (problem-solving and spatial relations) were significant: 82.4% of the variance for basic driving skills; 74.1% of the variance for overall driving skills. (b) Symbolic Representation Scale scores increased the predictive power to 87.1% for basic skills and 80.7% for overall mobility.	
Mountain et al., 2010	Baseline, Post training	*n* = 10 67.1 years (41–87) stroke neglect group *n* = 6 non-neglect group *n* = 4 the Sunny Brook Neglect Battery was used to determine neglect and non-neglect group	Neglect group Wheelchair skills training program, version 3.2 5 training sessions 30 min each Non-neglect group Wheelchair skills training program, version 3.2 5 training sessions 30 min each	cognition (baseline only) (a) Neurobehavioral Cognitive Status Examination (b) Sunny Brook Neglect Battery (c) Visual Scanning Tests (d) 22 clock drawing PMD capacity (a) WST	cognition (baseline only) (a) M = 86.9% (SD 11.6) (b) M = 14.5 (SD 20.3) (c) M = 43.8% (SD 42.7) (d) M = 12.6 (SD 2.2) PMD capacity (a) overall group: *p* = 0.002 * neglect group: *p* = 0.740 non-neglect group: data not shown no significant difference in the extend of improvement between the neglect and non-neglect group (*p* = 0.749)	III 4/5
**Cross Sectional Study Design (*n* = 3)**						
Butler et al., 1984	1 data collection time	*n* = 13 31.3 months (20–37) paraplegia, cerebral palsy, arthrogryposis congenita, osteogenesis imperfecta, spinal muscular atrophy, four-extremity limb deficiency, hypotonic quadriplegia	PMD training: 4-month observation period PMD introduced at home by parents Parents had to encourage the child to sit in the PMD several hours a day, to give them the opportunity to experiment the PMD in open spaces, to permit supervised play and to respect resistance to engage in further activity	cognition (a) Development profile II PMD capacity (a) cumulative hours to learn the 7 skills (b) days to learn the 7 skills	cognition (a) 12 children scored in the normal range 1 had higher level of intellectual functioning PMD capacity (a) M = 34.4 cumulative hours (b) M= 16.3 days	III 2/5
Massengale et al., 2005	Cross-sectional	*n* = 62 40.4 years (18–72) diverse diagnosis (e.g., spinal cord injury, cerebral palsy, post-polio syndrome, stroke, traumatic brain injury…)	n.a.	cognition (a) MVPT-R (b) TONI-3rd (c) WAIS-R (comprehension/picture completion/digit span) PMD capacity (a) Power Mobility Road Test (performance score and time to complete) (b) Power Wheelchair Screening Form	correlations between cognition and PMRT performance score (a) r = 0.591, *p* = 0.000 * (b) r = 0.392, *p* = 0.003 * (c) comprehension: r = 0.297, *p* = 0.026 */picturcompletion r = 0.418, *p* = 0.001 */digit span: r = 0.315, *p* = 0.018 * correlations between cognition and time required to complete the PMRT (a) r = −0.707, *p* = 0.000 * (b) r = −0.324, *p* = 0.012 * (c) comprehension: r = −0.306, *p* = 0.019 */picture completion r = −0.418, *p* = 0.001 */digit span: r = −0.258, *p* = 0.048 *	
Tefft et al., 1998	Cross-sectional	*n* = 26 n.r 20–36 months diverse diagnosis (e.g., arthrogryposis, congenital myopathy, quadriplegia, polio syndrome, spina bifida…)	Wheelchair mobility training program 6 sessions of 1 h	Cognition (a) PPWST (object permanence, problem solving, spatial relations) Capacity and performance (a) Powered mobility program evaluation	Cognition/capacity/performance spatial relations and problem solving were significant and accounted for 57% of the variance in Powered mobility program evaluation scores (R^2^ = 0.57, F = 14.37, *p* < 0.0001 *)	
**Single Subject Research Design (*n* = 2)**						
Dawson et al., 2003	A baseline B intervention A baseline B post	*n* = 2 p1: 67 years old p2: 70 years old stroke, unilateral neglect	PMD training: 2-weeks intervention 30 min every weekday based on right hemisphere activation approach	cognition (a) Behavioural Inattention Test (b) Star cancellation test (c) Baking tray task PMD capacity (a) number of left collisions recorded (b) time taken negotiating an obstacle	cognition Participant 1: (a) continued to demonstrate unilateral neglect (b) performed within the normal limit, no significant change comparing the 3 phases (c) trend for rightward spatial bias to normal limit, significant change between phase A1 and B1 *p* = 0.005 * Participant 2: (a) continued to demonstrate unilateral neglect (b) performed consistently below the normal limits, minimal significant change between phase A2 and B2 *p* = 0.043 * (c) rightward spatial bias was consistently seen PMD capacity *Participant 1:* (a) significant reduction in the left collisions *p* = 0.04 * (b) significant reduction in the time taken to complete the obstacle course *p* = 0.003 * Participant 2: (a) significant reduction in the left collisions *p* < 0.0001 * (b) significant reduction in the time taken to complete the obstacle course *p* = 0.0015 *	IV 3/5
Kenyon et al., 2018	A baseline B intervention A baseline B intervention 6 weeks follow-up	*n* = 1 3 years 2 months spastic quadriplegic, cerebral palsy, cortical visual impairment micro cephalic seizure disorder	Individualized PMD training: 4-week (repeated twice) 1 time per week for 45–60 min Development of basic power mobility skills after identification of motivational and reinforcement factors, participant-specific goals, creation of an engaging environment, adaptation of a custom-made control unit, and individualized verbal and physical prompts	cognition (a) Dimension of Mastery Questionnaire (b) PEDI-CAT (cognitive) PMD capacity (a) Wheelchair Skills Checklist (b) Assessment of Learning Power mobility (c) PEDI-CAT (mobility)	cognition (a) baseline 1: 3.36 (0.32) intervention 1: 3.84 (0.23) baseline 2: 3.84 (0.23) intervention 2: 3.58 (0.17) (b) intervention 1: 51 (1.59) intervention 2: 50 (1.75) follow-up: 51 (1.52) PMD capacity (a) intervention 1: 1/7 skills intervention 2: 5/7 skills follow-up: 5/7 skills (b) intervention 1: phase 2 (curious novice) intervention 2: phase 4 (advanced beginner follow-up: phase 5 (sophisticated beginner) (c) intervention 1: 38 (4.41) intervention 2: 47 (1.65) * follow-up: 49 (1.43) * * Standard error measure exceeded in the mobility domain	IV 4/5
**Case(s) Report (*n* = 5)**						
Benford et al., 2017	7 data collection time	*n* = 1 23 years old male spastic quadriplegic epilepsy profound and multiple learning disabilities	Driving to learn develop understanding of cause effect 23 weeks 31 sessions (during 30 to 45 min)	cognition (a) qualitative data (mood and fatigue) PMD capacity (a) Assessment of Learning Power mobility (a1) attention related to PMD use (a2) understanding related to PMD use (a3) expression emotions related to PMD use (a4) activity and movement	cognition (a) “*fatigue impacted performance*” “*his mood generally improve over the period that he participated in sessions*” PMD capacity (a1) pre: 3/post: 5 (a2) pre: 3/post: 5 (a3) pre: 3/post: 5 (a4) pre: 3/post: 5	IV 2/5
Jones et al., 2003	pretest 1 (beginning) pretest 2 (3 months) postest (6 months)	20 months old type II spinal muscular atrophy	intervention based on motor learning principles 6 weeks	cognition (a) PEDI (cognitive) (b) Battelle Developmental Inventory (b1) adaptive (b2) communication (b3) cognitive PMD capacity (a) PEDI (mobility)	cognition (a) positive trend (b) (b1) normal development (b2) positive trend (b3) age-equivalent scores increased greater than the period PMD capacity (a) positive trend	IV 4/5
Kenyon et al., 2015	pre post	*n* = 1 18 years old spastic quadriplegic cerebral palsy cortical visual impairment	PMD training 12 weeks 2 times a week 60 min Intervention based on practice power mobility skills and self-exploration within meaningful activities	cognition (a) Power mobility screen (cognition: judgement and abstraction) PMD capacity (b) Power mobility screen (motor scale)	cognition (a) pre: 9/21 post: 19/21 PMD capacity (b) pre: 16/30 post: 24/30	IV 4/5
Kenyon et al., 2017	case series baseline post	*n* = 3 n.a. P1:1 year 5 months P2: 2 years 5 months P3: 3 years 5 months cerebral palsy	PMD training: 12 weeks 1 time per week for 60 min Identification of individualized goals, practice within an engaging and playful environment	cognition (a) PEDI-CAT (cognitive) (b) Dimension of Mastery Questionnaire (cognitive persistence) PMD capacity (a) PEDI-CAT (mobility) (b) Assessment of Learning Power mobility	cognition scales score (SE) (a) P1: pre: 40(3.3)/post: 46(2.3) * P2: pre: 477(2.0)/post: 50(1.8) * P3: pre: 58(1.1)/post: 59(1.1) * (b) P1: stable pre-post (bellow the norm) P2: increase pre-post (bellow the norm) P3: stable pre-post (bellow the norm) PMD capacity (a) P1: pre: 41(3.7)/post: 43(3.2) P2: pre: 34(5.6)/post: 46(3.3) * P3: pre: 41(3.8)/post: 49(2.8) * (b) P1: pre: novice/post: advanced beginner P2: pre: novice/post: beginner P3:pre: level advanced-beginner/post: competent	IV 4/5
Lynch et al., 2009	case report	*n* = 1 7-month-old cerebral palsy	Infant-friendly training 5 months 3–4 time per week Experiences gained from the Directional Driving trials (DETAILS) and the open exploration period (DETAILS)	cognition (a) Balley III (a1) cognition (a2) language reception (a3) language expression PMD mobility: (a) joystick activations (b) path length (meters) (c) total path length (meters) (d) percent directed driving success	cognition (a) (a1) pre:7/post:49 (a2) pre:11/post: 16 (a3) pre: 9/post: 14 PMD mobility: (a) positive trend and stable (b) small positive trend (c) positive trend (d) 0 during the first training month/then positive trend	
Nilsson et al., 2003	case report baseline and 12 months follow-up	*n* = 2 n.a. P1: 5 years old P2: 4 years old profound cognitive disabilities, visual and motor impairment	intensive PMD training: 4 months 1–3 time per week 30–90 min Manual guidance and hand-over-hand assistance, verbal feedback and natural consequences were used to teach each child, verbal description of the activity. After 4 months the training was transferred to the children’s homes.	video-recordings: facial expressions, body movement, vocalizations, and reaction to interaction field notes: new and special reaction to behaviors in-depth interviews: how parents and assistants experienced the training and the children’s changing behaviors over time	P1: 1st session: “moving in circles”; “unable to place her hand on the joystick unaided”; “did not display any intention of doing so” follow-up: “after a great effort and with obvious intention, she was able to carry out self-initiated driving” P2: 1st session: “hand-over-hand driving”; “no sign of understanding that he could initiate the driving” follow-up: “able to keep the joystick in driving position”; “fluctuated between guided driving, unintentional driving, and driving after release of guidance”, “occasionally he managed to perform self-initiated driving”	

*, statistically significant; n.r., not reported; n.a. not applicable; BADS, Behavioural Assessment of the Dysexecutive Syndrome; CG, control group; IQ, Intelligence quotient; IG, intervention group; MoCA, Montreal cognitive assessment test; MVPT-R, Motor free Visual Perception Test-Revised; P, participant; PEDI, Paediatric Evaluation of Disability; PEDI-CAT, Paediatric Evaluation of Disability- Computer Adaptive Test; PPWST, Pediatric Powered Wheelchair Screening Test; PMC, Power Mobility Checklist; PMD, Powered Mobility Device; TONI-3^rd^, Test of Nonverbal Intelligence—3rd edition; WAIS-R, Wechsler Adults Intelligence Scale—Revised; WSP, Wheelchair Skills Program; WST, Wheelchair Skills Test.

**Table 2 ijerph-18-12467-t002:** Classification of outcome measures (and items) related to cognition used in the studies according to the International Classification of Functioning, Disability and Health (ICF).

Outcome Measures’ Name	Used In	Outcome Measure Brief Description	Specific Cognitive Function, Evaluated in Included Study, Classified According to the ICF
Addenbrooke’s Cognitive Examination (ACE-R) [34]	Cullen et al., (2008)	− screening neuropsychological test − 5 cognitive subscales (attention/orientation, memory, verbal fluency, language, visuospatial function) and the MMSE. − language subscale (i.e., naming and comprehension) was used	reception of spoken language (d16700), expression of spoken language (d16710)
Behavioural Assessment of the Dysexecutive Syndrome (key search subtest) (BADS) [35]	Cullen et al., (2008)	− predict everyday problems associated with cognitive impairment − 6 subtests (cognitive flexibility, problem solving, planning, judgement and estimation, and behavioral regulation). − problem solving subtest was used	problem solving (b1646)
Behavioural Inattention Test [36]	Cullen et al., (2008)	− screening tool evaluating the implications of the neglect deficit in everyday life − 6 subtests (line crossing, letter cancellation, star cancellation, figure and shape copying, line bisection, representational drawing). − line crossing subtest was used	visuospatial perception (b1565)
Depression Anxiety Stress Scale-21 (DASS-21) [37]	Cullen et al., (2008)	− self-report questionnaire about depression, anxiety, and stress − 21-item	psychic stability (b1263)
F-A-S test [38]	Cullen et al., (2008)	− subtest of the Neurosensory Center Comprehensive Examination for Aphasia measure of verbal fluency. − 1 min to give as many words as possible beginning with a letter	expression of spoken language (d16710)
Motor Free Visual Perception Test-Revises (MVPT-R) [39]	Massengale et al., (2005)	− standardized measure of visual perceptual skills − 65 perceptual tasks	visuospatial perception (b1565)
Pediatric Powered Wheelchair Screening Test (PPWST) [26]	Furumasu et al., (2004) and Tefft et al., (1999)	− evaluate young child’s cognitive readiness to operate a powered wheelchair − subscales (level of development across problem solving and spatial relations scales)	visuospatial perception (b1565), organization and planning (b1641) problem solving (b1646)
Repeatable Battery for the Assessment of Neuropsychological Status (RBANS) [40]	Cullen et al., (2008)	− neuropsychological battery − 5 indexes (immediate and delayed memory, attention, visuospatial perception and language)	short term memory (b1440), long term memory (b1441), attention functions (b140), visuospatial perception (b1565), mental function of language (b167)
Road Map Test of Direction Sense test [41]	Cullen et al., (2008)	− assess topographic orientation − present a street map on which are drawn 2 routes taken by a hypothetical traveler. The subject has to imagine himself travelling along the specified route and to spatially rotate	cognitive function orientation to place (b1141), cognitive flexibility (b1643)
Symbolic Representation Scale [20]	Furumasu et al., (2004)	− developed specifically for the study − assess child’s understanding of symbols	integrative language functions (b1672)
Test Of Nonverbal Intelligence—3rd edition (TONI-3rd) (Brown et al., 1996)	Massengale et al., (2005)	− evaluate abstract reasoning, problem solving, aptitude and intelligence for individuals with language difficulties and sensory deficits	abstraction (b1640), problem solving (b1646), intellectual function (b117)
Visual Object and Space perception Battery (VOSP) (Lezak et al., 2004)	Cullen et al., (2008)	− battery of 8 tests − assess object or space perception	visuospatial perception (b1565)
Wechsler Adults Intelligence Scale—Revised (WAIS-R) (Matarazzo, 1996)	Massengale et al., (2005)	− assess aspects of intelligence − 8 subtests (digit span, attention, concentration, memory, comprehension, judgement, reasoning skills, picture completion)	sustaining attention (b1400), short term memory (b1440), judgement (b1645)

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
