# Peer review of "Relationships between Cognitive Functioning and Powered Mobility Device Use: A Scoping Review"

_ijerph, 2021, doi:10.3390/ijerph182312467_

Round 1

Reviewer 1 Report

Expand the methods part of your Abstract. 

Rewrite explaining better lines 82-85. Why "was more appropriate to conduct a scoping review than a systematic review"? Make a better argument why you used a scoping review. 

One concern. Some studies are old and some more recent, the span is almost 40 years (1984 - 2017). PMDs changed in the last 40 years. Can we consider them the same devices?

You say on line 448 "In clinical practices it appears primordial to consider cognitive functioning when providing a PMD." But some of your studies are old and the practices are changing. Please address this issue.

In your implications section, you should discuss (or add a discussion section) what are your recommendation from the practical aspect of the issue. 

You conclude that "... high variability in the designs, outcome tools and PMD trainings used through the studies..." How can you then draw conclusions? Give a better recommendation that further studies are needed.

Author Response

Manuscript ijerph-1468164

Response to reviewers

November, 22nd 2021

Dear reviewers,

Thank you for your review.

Comments and remarks helped to improve quality of our manuscript. You will find our responses and actions in the table below. Modifications in the manuscript are highlighted in grey.

Kindest regards,

Krista Best

Reviewer 1

Comment : Rewrite explaining better lines 82-85. Why "was more appropriate to conduct a scoping review than a systematic review"? Make a better argument why you used a scoping review. 

Response: Explanation why authors used a scoping review was rephrased and an additional citation was added to justify the use of a scoping review over a systematic review: Munn, Z., Peters, M.D.J., Stern, C. et al. Systematic review or scoping review? Guidance for authors when choosing between a systematic or scoping review approach. BMC Med Res Methodol 18, 143 (2018). https://doi.org/10.1186/s12874-018-0611-x

Action: line 82 - 88

“A scoping review was conducted to explore the nature and the extent of research evidence [14,15] and to clarify concepts related to cognitive function and PMD use. The intent of the research was to provide a scope of current evidence [16] that included all study designs and all ages of power wheelchair users. Findings from this scoping review may orient research question development and selection of inclusion criteria for a future systematic review [15,16].

Comment : One concern. Some studies are old and some more recent, the span is almost 40 years (1984 - 2017). PMDs changed in the last 40 years. Can we consider them the same devices?

Response 1: Authors recognized the relevance of this comment. Thoughts were added in the discussion section.

Action 1: line 475 - 486

“Exploring the relationship between PMD use and cognitive functioning should also consider PMD technological evolutions. The studies included were conducted between 1984 and 2017 and PMD have greatly changed over time. Still, cognitive functioning implied in PMD use remains the same. Moreover, such research may orient the development of intelligent PMD which have similar rationale providing mobility options for people with dual motor and cognitive impairment [45,46]. For example, intelligent PMD may enhance secure driving (i.e., automatic stop) and can facilitate some complex maneuver (i.e., enter in an elevator).”

Response 2: The large range in publications dates and in relation to the evolution of PMD design has also been added as a limitation.

Action 2: line 494-495

“Second, in relation to the evolution of PMD and clinical practice, the large range in publication dates represents a limitation.”

Comment : You say on line 448 "In clinical practices it appears primordial to consider cognitive functioning when providing a PMD." But some of your studies are old and the practices are changing. Please address this issue.

Response 1: Authors changed the phrase to address the fact that some studies are old.

Action 1: line 450-455

“Despite high variability in study designs and publication dates, it appears primordial to consider cognitive functioning when providing a PMD. Although there have been changes in clinical practice and evolution of PMD

the role of cognitive functioning is still important for PMD use. While it is true that some intelligent PMD may play a role in reducing cognitive load for some individuals, artificial intelligence cannot assume all responsibility for the person.”

Response 2: Consideration of the evolution of clinical practices has also been added as a limitation.

Action 2: line 494-495

“Second, in relation to the evolution of PMD and clinical practice, the large range in publication dates represents a limitation.”

Comment : In your implications section, you should discuss (or add a discussion section) what are your recommendations from the practical aspect of the issue. 

Response: Practical recommendations were added in the section implication. 

Action: line 468 - 471

“Innovative training approaches customized to individuals with cognitive impairment could, for example, include practicing meaningful activities with the PMD and be conducted in realistic environments and contexts for the user. Meaningful activities should be client-centered and driven largely by client goals.”

Comment : You conclude that "... high variability in the designs, outcome tools and PMD trainings used through the studies..." How can you then draw conclusions? Give a better recommendation that further studies are needed.

Response: Specifications about the required studies have been added in the end of the conclusion.

Action: line 522 - 525

Such studies are required to explore how cognitive functioning and psychological factors, such as confidence, influence PMD use, and to document which cognitive evaluations could be used with individuals with dual motor and cognitive impairment in clinical practices.”

Reviewer 2 Report

This work explores the relationships between cognitive functioning and Powered Mobility Devices use among users with dual cognitive and mobility impairment. The objectives were to identify cognitive functions necessary to use a Powered Mobility Devices and describe available training approaches. Findings suggests that cognitive functioning should not be overlooked when providing Powered Mobility Devices. However, given that individuals with heterogeneous cognitive impairment can improve their capacities, cognitive impairment should not preclude Powered Mobility Devices provision.

The research was carefully designed, the conclusions are supported by the results, and the provided information is relevant for the knowledge field. Nevertheless, some issues could be addressed before this manuscript could be considered for publication.

1 This work contributes to a fundamental advancement of knowledge to ensure best practices for Powered Mobility Devices provision, and best practices for Powered Mobility Devices development.

2 The Conclusion section should provide detailed recommendations for future work summarizing all the mentioned in the Discussion section.

3 For new Powered Mobility Devices it could be interesting to know the opinion of the authors regarding the inclusion of Artificial Intelligence features to compensate for some cognitive impairments.

Author Response

Manuscript ijerph-1468164

Response to reviewers

November, 22nd 2021

Dear reviewers,

Thank you for your review.

Comments and remarks helped to improve quality of our manuscript.

You will find our responses and actions in the table below. Modifications in the manuscript are highlighted in grey.

Kindest regards,

Krista Best

Reviewer 2

Comment : The Conclusion section should provide detailed recommendations for future work summarizing all the mentioned in the Discussion section.

Response: Recommendations related to the development of PMD training approach has been added in the conclusion.

Action: line 515 - 518

“To improve mobility and quality of life for individuals with dual motor and cognitive impairment innovative training approaches should be developed. Such training approaches could include practice of client-centered goals related to using their PMD and be conducted in the users’ actual environment.”

Comment: For new Powered Mobility Devices, it could be interesting to know the opinion of the authors regarding the inclusion of Artificial Intelligence features to compensate for some cognitive impairments.

Response: Considerations related to intelligent power mobility devices have been added in the discussions section.

Action: line 478 - 487

“Moreover, such research may orient the development of intelligent PMD which have similar rationale providing mobility options for people with dual motor and cognitive impairment [45,46]. For example, intelligent PMD may enhance secure driving (i.e., automatic stop) and can facilitate some complex maneuver (i.e., enter in an elevator). While there may be added benefits to PMD users and caregivers (e.g., less worry and risk of accidents), controlled mobility can limit the individual’s ability to make choices about their participation in meaningful activities. Properly programing an intelligent PMD to the users’ cognitive abilities could enhance mobility without restricting autonomy and choice [11].
